# Novel Long-Acting Oxytocin Analog with Increased Efficacy in Reducing Food Intake and Body Weight

**DOI:** 10.3390/ijms231911249

**Published:** 2022-09-24

**Authors:** Clinton T. Elfers, James E. Blevins, Therese S. Salameh, Elizabeth A. Lawson, David Silva, Alex Kiselyov, Christian L. Roth

**Affiliations:** 1Center for Integrative Brain Research, Seattle Children’s Research Institute, 1900 Ninth Avenue, Seattle, WA 98101, USA; 2VA Puget Sound Health Care System, Office of Research and Development Medical Research Service, Department of Veterans Affairs Medical Center, Seattle, WA 98108, USA; 3Division of Metabolism, Endocrinology and Nutrition, Department of Medicine, University of Washington School of Medicine, Seattle, WA 98195, USA; 4Neuroendocrine Unit, Massachusetts General Hospital, Boston, MA 02114, USA; 5Department of Medicine, Harvard Medical School, Boston, MA 02114, USA; 6Mass General Brigham, One Main Street, Suite 510, Cambridge, MA 02142, USA; 7Myocea, Inc., 9833 Pacific Heights Blvd, San Diego, CA 92121, USA; 8Division of Endocrinology, Department of Pediatrics, University of Washington, Seattle, WA 98105, USA

**Keywords:** obesity, oxytocin, oxytocin agonist, in vivo stability, pharmacology, diet-induced obese rats, food intake, body weight, brown and white adipose tissue

## Abstract

Oxytocin (OXT) analogues have been designed to overcome the limitation of the short half-life of the native OXT peptide. Here, we tested ASK2131 on obesity related outcomes in diet-induced obese (DIO) Sprague Dawley rats. In vitro function assays were conducted. The effects of daily subcutaneous injections of ASK2131 vs. OXT and pair-feeding were assessed on food intake and body weight in vivo. ASK2131 is a longer-lasting OXT analog with improved pharmacokinetics compared to OXT (T_1/2_: 2.3 vs. 0.12 h). In chronic 22-day administration, ASK2131 was administered at 50 nmol/kg, while OXT doses were titrated up to 600 nmol/kg because OXT appeared to be less effective at reducing energy intake relative to ASK2131 at equimolar doses. After 22 days, vehicle-treated animals gained 4.5% body weight, OXT rats maintained their body weight, while those treated with ASK2131 declined in weight continuously over the 22-day period, leading to a 6.6 ± 1.3% reduction (mean ± standard error) compared to baseline. Compared to their pair-fed counterparts, ASK2131-treated rats showed a more pronounced reduction in body weight through most of the study. In summary, ASK2131 is a promising OXT-based therapeutic, with extended in vivo stability and improved potency leading to a profound reduction in body weight partly explained by reduced food intake.

## 1. Introduction

Obesity and its comorbidities cause significant, yet preventable, morbidity and mortality and is a major public health burden worldwide [1]. Diet and exercise alone are not sufficient to produce long term decreases in weight, as many studies have demonstrated weight regain after one year [2]. Although drug interventions can improve weight loss, treatments often have limited long-term efficacy and are associated with safety and tolerability issues with little or no impact on disease reversal [3].

Oxytocin (OXT) has emerged as an attractive target for treating obesity. There is a growing body of evidence that peripheral administration of OXT reduces food intake, enhances energy expenditure, and prevents or reduces weight gain in obese rodent models and non-human primates. This is thought to occur despite the reduced ability of OXT to penetrate into the brain [4] from the periphery, although small amounts have been detected in the CSF [5,6]. Instead, the mechanism of OXT-related reductions in food intake is thought to occur, in part, through activation of the vagal afferent neurons [7,8]. In humans, the use of intranasal administration as a non-invasive approach to bypass the blood–brain barrier has become increasingly popular in the treatment of diseases [9]. Few data are published demonstrating a reduction of body weight in response to intranasal OXT administration in humans with obesity [10,11]. However, multiple administrations four times daily might be required [10], making adherence to drug regimens often difficult and long-term effects are not well established in humans. In addition, potential crossover of OXT to vasopressin receptors is of concern given that the weight loss effects of intranasal OXT are seen at supraphysiologic doses, raising the possibility of cardiovascular effects (e.g., tachycardia and heart failure via vasopressin receptor 1a [V1aR]), hyponatremia (via vasopressin receptor 2 [V2R]) [12], and, upon potential CNS exposure, anxiety and aggression in at risk patients (V1aR, vasopressin receptor 1b [V1bR]) [13,14,15]. Selective OXT analogs are a promising therapeutic avenue for obesity as they could provide a longer duration of action to increase efficacy and selectivity to the OXT receptor to avoid unwanted off-target effects.

Our group has previously shown that ASK1476, a long-lasting, peripherally acting, potent and selective OXT analog, is able to effectively reduce food intake and body weight in comparison to native OXT [16]. This OXT analog is a full agonist at OXT receptor, and it addressed some of the limitations of using OXT as a weight loss therapeutic. In particular, it increased the short half-life of OXT from ∼5 min to 3.2 h in vivo, did not show activity at the V1aR which was an improvement compared to the V1a receptor agonistic profile of OXT [17,18], and resulted in mild toxicity at high doses, e.g., malaise [16].

In this study, we report the design, pharmacokinetics, receptor selectivity of a novel oxytocin agonist, ASK2131, and its efficacy in reducing caloric intake and body weight.

## 2. Results

### 2.1. In Vitro Functional Assay

The design of ASK2131 was based on recent reports describing OXT receptor (OXTR) agonists with extended in vivo stability [17,18]. Specifically, we prepared carba hexapeptide core by replacing one S atom of the disulfide bridge with respective CH2 group [17]. Pro7 was changed to Gly7 to amend functional activity of the resulting molecule at the vasopressin V1a receptor (V1aR) while maintaining its OXTR agonistic potency [17]. Leu8 was replaced with Lys8 and the terminal NH group of the Lys8 was lipidated [18] to enhance binding of the molecule to plasma proteins in vivo (Figure 1). This was expected to reduce the free fraction of ASK2131 available for enzymatic degradation. Receptor pharmacology profiles of both OXT and ASK2131 against OXTR, V1aR and V2R are summarized in Table 1.

Notably, in our hands ASK2131 did not display V1aR activation whereas its agonist activity for OXTR was 1.1 nM [17]. Furthermore, the molecule was a potent activator of V2R showing EC_50_ of 0.36 nM in the cAMP assay.

### 2.2. Pharmacokinetics

Pharmacokinetics data for OXT and ASK2131 following subcutaneous (sc) administration to rats are summarized in Table 2. Following sc administration in rats, the lipidated analogue ASK2131 exhibited considerably reduced plasma clearance (CLplasma = 1.2 mL/mg/kg) and an increased steady-state volume of distribution (Vd = 0.80 L/kg) compared to OXT, resulting in a half-life (T_1/2_) of 2.3 h and T_max_ of 4 h. The lipidated analog ASK2131 also displayed a total maximal plasma concentration (Cmax) of 276 nM and free Cmax of 0.6 nM at 4 h post-dose further suggesting increased plasma binding profile of the molecule. The area under the concentration-time curve from dosing (time 0) to the time of the last measured concentration (AUClast) was 1.95 h·mg/mL suggesting good plasma stability and relevant exposure for subsequent efficacy studies.

### 2.3. Body Weight, Food Intake and Metabolic Outcomes in Long-Term Study

#### 2.3.1. Changes in Body Weight (Main Outcome)

After 2 days of drug administration, ASK2131-treated animals showed a significant decrease in body weight compared to the vehicle-treated group (predicted mean difference (PMD) −3.83%; 95% CI −6.52 to −1.15%; *p* = 0.0011) and the OXT-treated group (PMD −3.77%; 95% CI −6.45 to −1.08%; *p* = 0.0014). This was maintained until study completion leading to a body weight reduction of 11.5% over vehicle treatment (Figure 2A–C). On treatment day 9 of the study, after 8 days of drug administration, OXT-treated animals showed a significant difference in body weight compared to the vehicle-treated group (PMD −3.12%; 95% CI −5.80 to −0.43%; *p* = 0.014) leading to a difference in weight change of 5% weight loss vs. vehicle at the end of treatment. Pair-fed ASK2131 treated animals were given the same amount of food as their paired ASK2131 treated animal. At treatment day 20 of the study, after 19 days of pair feeding, the ASK2131 pair-fed group showed significant body weight reduction compared to the ASK2131-treated group (PMD −2.90%; 95% CI −5.58 to −0.21%; *p* = 0.027) (Figure 2A).

#### 2.3.2. Changes in Food Intake

Analysis of food intake pattern showed a consistent and stronger reduction of food intake starting at day 1 of treatment compared to OXT with comparable doses (50 nmol/kg/d) (Figure 2D and Figure 3A) with a 39 ± 9% reduction of food intake at day 3 compared to baseline and pronounced reduction of food intake compared vehicle-treated group (at day 3: PMD −45.99%; 95% CI −80.47 to −11.51%; *p* = 0.0102). Even when doses were increased for OXT every 2 days up to 600 nmol/kg while maintaining the dose for ASK2131 at 50 nmol/kg/d, food intake reduction was more pronounced in response to ASK2131 vs. OXT as demonstrated by daily food intake (Figure 2D) and cumulative food intake (Figure 3B). OXT-animals did not exhibit a significant reduction in food intake compared to the vehicle-treated group.

No groups exhibited either significant changes in water intake compared to baseline or significant differences in cumulative water intake relative to other treatment groups (Figure 2E).

#### 2.3.3. Metabolic Outcomes

At the time of euthanasia following 22 days of treatment, blood samples were collected through cardiac puncture. Cholesterol was significantly reduced in ASK2131 treated animals and their pair-fed controls vs. vehicle treated animals. There were no significant between-group differences in levels of glucose, insulin, leptin, adiponectin, high-density lipoprotein (HDL), calculated low-density lipoprotein (LDL), triglycerides, alanine transaminase (ALT), or aspartate transaminase (AST) (Table 3).

#### 2.3.4. Gene Expression

While the relatively small n in this study made determining the effect of OXT or ASK2131 on adipose tissue thermogenesis difficult, there was a trend towards elevated levels of UCP1 (mean difference (MD) 3.02, 95% CI 1.12 to 4.93, *p* = 0.001) and Cox8b (mean rank difference (MRD) 9.42, *p* = 0.055) in inguinal white adipose tissue (IWAT) in response to OXT. In comparison to the vehicle treated group, none of the examined genes showed a significant response in the ASK2131 group. The level of expression in IWAT and interscapular brown adipose tissue (IBAT) for the thermogenic marker UCP1 was lower in response to ASK2131 treatment relative to OXT treatment (IWAT: MD −2.32, 95% CI −4.23 to 0.42, *p* = 0.014; IBAT: MRD −12.5, *p* = 0.011). Appendix A shows results for UCP1 (A, B) and Cox8b (C, D).

#### 2.3.5. Secondary Analysis of OXT vs. ASK1476 vs. ASK2131

Direct comparison of the long-term efficacy of ASK2131 vs. ASK1476 vs. OXT, using data from our current and prior [16] experiments is not without notable limitations given differences in experimental design. After 22 days of treatment, the effects of treatment on cumulative food intake relative to vehicle treated controls were comparable for ASK1476 (30–120 nmol/kg/d) and ASK2131 (50 nmol/kg/d) treated animals (ASK2131 vs. Vehicle: MD −621.9 kcal, 95% CI −1031 to −213.1, *p* < 0.0008; ASK1476 vs. Vehicle: MD −484.3 kcal, 95% CI −838.3 to −130.3, *p* = 0.0029), and while body weight reduction was greater in ASK2131 vs. ASK1476 treated animals relative to baseline (ASK2131: −6.6 ± 1.3%, ASK1476: −1.7 ± 3.6%, *p* = 0.006), their mean effect of treatment when compared to their respective vehicle controls were comparable (ASK2131 vs. Vehicle: MD −11.5%, 95% CI −15.6 to −7.4, *p* < 0.0001; ASK1476 vs. Vehicle: MD −10.2%, 95% CI −13.8 to −6.7, *p* < 0.0001). The optimal dosing strategy for OXT was the one used in the ASK1476 experiment (600 nmol/kg/d) yielding reductions in food intake relative to vehicle (MD −471.3 kcal, 95% CI −825.3 to −117.3, *p* = 0.0039) comparable to ASK1476 and ASK2131; body weight change relative to baseline (−3.49 ± 1.0%) were not significantly different from either ASK1476 or ASK2131.

### 2.4. Body Weight and Food Intake in Dose Escalation Study

To test the range of responses to different drug doses, diet-induced obese (DIO) Sprague Dawley rats were administered once daily doses of ASK2131 increasing every two days starting at 1 nmol/kg and ending at 100 nmol/kg. Body weight showed continuous decline over the course of the treatment period and on average each rat showed an 8.4% loss in body weight compared to baseline (Figure 4A). Food intake reduction showed a U-shaped response. The highest level of reduction (~55%) was observed at 2 nmol/kg. At doses < 2 nmol/kg and >5 nmol/kg, 2-day averaged food intake was constrained to a 20–25% reduction relative to baseline (Figure 4B). Water intake showed a modest decline with treatment onset but was not significantly different from baseline with increased dosing of ASK2131 (Figure 4C).

During the course of this study, one of the DIO rats had an adverse event and was terminated before study completion. This rat developed severe diarrhea and experienced rapid weight loss. Necropsy of this animal revealed enteritis in the small intestine, mildly enlarged liver and spleen, and a focal dermatitis lesion over the left hip, likely an injection site wound. This rat was excluded from analysis as this was an isolated event which we could not wholly determine to be a result of the ASK2131 treatment. No other animals treated with ASK2131 showed any signs of an adverse response.

## 3. Discussion

In this series of experiments, we report superior efficacy of a novel selective OXT agonist in comparison to synthetic OXT in DIO rats, only partly explained by reduced food intake. Our group recently published a paper demonstrating that ASK1476, another injectable long-acting selective OXT receptor agonist, resulted in similar reduction in food intake and body weight at lower doses to synthetic OXT in DIO rats [16]. ASK1476 [PF-06655075 or PF1 [17] was modified to increase half-life and plasma protein binding compared to native OXT. Pharmacokinetic data suggested that the modifications leading to the development of ASK1476 were successful in improving its in vivo stability. The development of ASK1476 showed promise as an OXT based anti-obesity drug intervention, since this once daily subcutaneous administration reduced food intake body weight comparably to human studies which currently require 4× daily intranasal administration [10]. However, for ASK1476 we observed some adverse effects occurring at higher doses of drug administration and indication of malaise [16], signifying room for optimization of this peptide.

In this current study, we tested a novel selective OXT receptor analog, ASK2131 for its effect on food intake and weight gain in DIO Sprague Dawley rats. In vitro function assays were conducted to assess its ability to bind to the oxytocin and vasopressin (V1aR and V2R) receptors.

Neither molecule, ASK1476 or ASK2131, is active at the V1aR, but they both activate V2R. Nevertheless, in all our longitudinal in vivo studies, ASK2131 was well tolerated without relevant changes of water intake. While water intake is a reasonable initial assessment, further experiments investigating possible fluid imbalance are warranted. In addition, pharmacokinetics of this peptide were examined following subcutaneous injection. An improved, i.e., longer, plasma half-life was measured after peripheral injection compared to OXT which will help to reduce dosing frequency.

For the long-term study, the dose of 50 nmol/kg was chosen based on previous pilot experiments. While this dose demonstrates an average 28% decline in food intake during the whole 22-day treatment, the first 8 days of treatment showed a 35% decrease, and the last 8 days showed a 25% reduction in food intake. Comparison of ASK2131-treated rats to their pair fed counterparts demonstrated that the weight loss due to ASK2131 was not driven solely by the reduction in food intake. OXT-treated animals showed a significant difference in body weight compared to the vehicle-treated group leading to a difference in weight change of 5% weight loss vs. vehicle at the end of treatment. This is more likely due to the vehicle-treated animals’ continuous weight gain, than OXT treatment causing weight decline.

In our study, the lack of stimulation of uncoupling protein-1 (UCP1) and Cox8b mRNA levels in IWAT and IBAT might indicate that ASK2131 does not have a major impact on stimulation of thermogenesis by these molecular mechanisms in contrast to OXT. Studies in non-human primates have shown that OXT elicits weight loss through increases in energy expenditure [19]. Conversely, reduced OXT signaling is associated with obesity and reduced energy expenditure and thermogenesis in brown adipose tissue (BAT) [20,21]. The exact mechanism by which this occurs is unknown. Results from our exploratory gene expression study raise the possibility that systemic OXT might elevate the gene expression of UCP-1 in IWAT of DIO rats. This finding is consistent with earlier work from Plante and colleagues who reported that systemic minipump infusions of OXT (~66.2 nmol/day) appeared to increase UCP-1 content in subcutaneous, perirenal and epicardiac fat of Leprdb mice [22]. Consistent with these findings, Yuan and colleagues also found that systemic OXT (100 nmol/day) increased UCP-1 mRNA and protein in IWAT in DIO mice [23]. In contrast to our findings in DIO rats, Yuan found that systemic oxytocin also elevated UCP-1 mRNA and protein in in IBAT in DIO mice [23]. Whether this is due, in part, to species differences or dosing remains to be determined.

We also acknowledge that one limitation to the gene expression analyses was the relatively small group size, and therefore these results need to be interpretated with caution. This may have made it more difficult to measure more subtle changes in thermogenic gene expression in response to systemic OXT and ASK2131 in both IWAT and IBAT. Future studies with increased N/group will be helpful in order to make more definitive conclusions about the effects of systemic OXT and ASK2131 on thermogenic gene expression in IBAT and IWAT.

To test the full range of dose responsiveness, a dose escalation study was completed on DIO Sprague Dawley rats, as it has been reported that obese rodents appear to be more sensitive to OXT than their lean counter parts [24,25]. In addition, the effect of long-term exposure to ASK2131 was examined by treating DIO Sprague Dawley rats for 22 days with this peptide in comparison to native OXT. At the low dose of 2 nmol/kg (0.0011 mg), we observed a 47% decrease in food intake in the DIO Sprague Dawley rats compared to their baseline food intake. To our knowledge, this is the most profound food intake reduction observed by an OXT analog to date. The dose response curve to ASK2131 was U-shaped. This U-shaped curve was evident in both the lean and DIO dose escalation studies. At 2 nmol/kg, rats had already shown an ~4% decrease in body weight compared to baseline after only four treatments. This is of particular note when compared to studies examining the effect of OXT on food intake and body weight. In a study using non-human primates, a 3.3% change in body weight was observed after one month of treatment with OXT, two-weeks at 0.2 mg/kg followed by two-weeks at 0.4 mg/kg [19]. In DIO rats, a single injection of 1000 μg (1000 nmol) of OXT led to a 12% decrease in food intake and a modest 0.01% change in body weight after a single dose [24]. These doses, up to 500× the dose of ASK2131 used, had modest, yet significant effects on food intake and body weight. The U-shaped response we observed has also been reported in human trials using intranasal OXT. For example, a study by Wynn et al. tested doses from 8 to 84 IU to determine the correct dose to use in the treatment of schizophrenia and determined that doses in the middle (36–48 IU) were most efficacious [26]. Furthermore, our results confirm the U-shaped response which we found in our prior study testing the OXT analogue ASK1476 [16]. This phenomenon is likely a result of OXT receptor internalization. Prolonged OXT treatment has been shown to promote desensitization of OXT receptors in vitro [27] and reductions of OXTR binding [28,29,30]. Alternatively, it is possible that at higher doses, OXT may have lower weight loss efficacy due to opposing actions at the V1bR. Activation of the V1bR increases ACTH release, which may in turn increase drive to eat and body, weight and single nucleotide polymorphisms of V1bR have been linked to higher body mass index [31]. In addition, U-shaped responses were seen in response to single OXT doses. In a neurobehavioral study studying acute fear responses, the OXT-induced inhibition of amygdala responses to fear revealed a significant decrease in amygdala activation assessed by functional neuroimaging following 24 IU doses, but not lower (12 IU) or higher (48 IU) doses [13].

The peptide ASK2131 was designed to further improve on ASK1476. Modifications to this peptide include replacement of the disulfide bridge with a thioether isostere and elimination of the Cys amino group. In addition, 7Pro was replaced with 7Gly to improve V1aR selectivity and 8Leu was changed to 8Lys followed by modification of terminal NH2 with polyethylene glycol spacer and a lipid palmitoyl group. The latter pharmacophore was shown by our team [16] and others [17] to enhance plasma protein binding of the resulting molecule and decrease the free fraction of the molecule available for metabolism, leading to the overall increase of ASK2131 half-life in vivo. Receptor pharmacology and pharmacokinetics for ASK2131 compared to ASK1476 and OXT do not indicate that ASK2131 should display enhanced performance. It should be noted that ASK2131 is still a potent activator of V2R, with an EC_50_ of 0.36 nM in the cAMP assay. While we did not observe any adverse effects caused by the activation of V2R throughout our longitudinal efficacy studies in rats, we did not test for potential side effects such as renal water retention and hyponatremia. The replacement of the disulfide bridge with a thioether isostere is a promising strategy used to improve the stability of disulfide-rich polypeptides [32]. This improved stability is potentially responsible for the improved function of ASK2131. Additionally, while ASK1476 was designed to be non-brain penetrating [17] to focus on a peripherally acting OXT receptor analog, it is unknown whether ASK2131 crosses the blood–brain barrier. Early work demonstrated that OXT does not cross the blood–brain barrier [4], instead it is effluxed from the brain into the periphery using PTS-1 [33]. Recent evidence suggests a receptor for advanced glycation end-products (RAGE) mediated mechanism of OXT transport to the brain [34]. Obesity is known to increase expression of RAGE [35,36], and it is possible that in our DIO Sprague Dawley rats increased transport which needs to be investigated for ASK2131 vs. OXT in a future study.

A secondary analysis comparing the ASK2131 efficacy data presented here with data from prior experiments by our team examining OXT and ASK1476 [16] indicated no notable differences between the three peptides with regard to their effects on food intake and body weight change. When interpreting these results there are a few limitations to consider, the most notable being differences in dosing strategies. The ASK1476 experiment utilized data from dose escalation studies to target an initial 20% reduction in caloric consumption and then adjusted doses to maintain that deficit. Concerns that doses of ASK1476 higher than 50 nmol/kg did not offer notably improved efficacy in either the dose escalation or long-term experiments led us to fix the ASK2131 dose at 50 nmol/kg in the current long-term experiment. Additionally, while the 50 nmol/kg dose used in the 22-day ASK2131 experiment was based on prior pilot experiment, our subsequent dose escalation study presented here indicated that ASK2131 may be more efficacious at lower doses. It is with the dose escalation experiments that the most notable difference in ASK2131 and ASK1476 are observed. The greatest anorectic effect achieved with ASK1476 occurred at the 300 nmol/kg/d dose yielding a ~34% reduction in caloric intake [16] while ASK2131 dosed at 2 nmol/kg/d yielded a ~55% reduction in caloric intake. An even more start contrast can be seen when we consider OXT which reached a maximum effect at 1200 nmol/kg/d yielding a ~36% reduction in caloric intake [16].

While this paper demonstrates strong reductions of food intake and body weight in response to a novel OXT receptor analog, there are limitations that need to be mentioned. First, these studies were performed in male animals only. Prior research showed reduction of food intake in response to OXT treatment also in female animals, but this needs to be established also for the novel peptide ASK2131 [25,37,38]. Second, our study did not include effects on energy expenditure in vivo, for instance by using indirect calorimetry and/or telemetry to assess changes in respiratory quotient, ambulatory activity, or thermogenesis. Additionally, the number of studies animals were relatively low, in particular for getting meaningful results for gene expression. While we plan to address these points in a larger follow-up study, we believe that the data show important improvements compared to the effects of native OXT peptide. In addition, more detailed dose-range, long-term toxicity studies focusing on in vivo V2R activation are in progress to evaluate potential side effects of ASK2131.

In summary, OXT-based therapeutics show promise as obesity medications. Here, we have shown data using a long-acting and potent OXT analog which has demonstrated profound reductions in body weight which can be partly explained by reduction in food intake and potentially also by its longer half-life and improved OXTR receptor agonist activity. Understanding dose response to novel OXT analogs that are more potent than the native OXT peptide will guide treatment of patients.

## 4. Materials and Methods

All procedures performed in rats were approved by the Institutional Animal Care and Use Committee at the Seattle Children’s Research Institute and were in accordance with the NIH Guide for Care and Use of Laboratory Animals. This facility is approved by the Association of the Assessment and Accreditation of Laboratory Animal Care International (AAALAC). For studies utilizing a DIO rat model, male Sprague Dawley rats (CD-IGS rats, strain 001; approximately 4 weeks of age; 51–75 g) purchased from Charles River Laboratories (Wilmington, MA, USA). These rats were fed a diet of 60% calories from fat (Formula: D12492; Research Diets, Inc., New Brunswick, NJ, USA; 5.21 kcal/g) for 4.5 to 6 months prior to the start of the study to induce obesity. For studies utilizing lean rats, male Sprague Dawley rats (CD-IGS rats, strain 001; 226–250 g) purchased from Charles River Laboratories (Wilmington, MA, USA). All rats had ad libitum access to food and water and were kept on a 12 h light/12 h dark cycle. Animals were individually housed in a temperature (22 ± 1 °C) and humidity (57 ± 4%) controlled room. All body weight measurements were taken just prior to the start of the dark cycle.

### 4.1. Design of ASK2131

The design of OXT analog ASK2131 was based on several published observations as well as on our recent encouraging in vivo data for its analog ASK1476 [PF-06655075 or PF1 [17]. To improve OXTR agonist activity for the targeted molecule, enhance vasopressin receptor (V1aR) selectivity and to increase its in vivo stability we introduced 2 key modification into ASK2131 (Figure 1). First, the disulfide bridge -S-S- of ASK1476 was replaced with the respective thioether isostere and the Cys amino group was eliminated, as described in Wisniewski et al. [18]. Secondly, 7Pro was replaced with 7Gly to improve V1aR selectivity and 8Leu was changed to 8Lys followed by modification of terminal NH2 with polyethylene glycol spacer and a lipid palmitoyl group.

### 4.2. Chemical Synthesis of Test Articles

OXT was purchased from Sigma-Aldrich for use in in vitro functional studies (St. Louis, MO, USA). The title compound ASK2131 was synthesized by a hybrid Boc/Fmoc strategy on MBHA resin using approach described in [18]. The following derivatives were employed: Boc-Gly OH, FmocHcy((CH_2_)_2_COOtBu)-OH, Fmoc-Asn(Trt)-OH, Fmoc-Gln(Trt)-OH, Fmoc-Ile-OH and Boc-Tyr(tBu)-OH or Boc-Phe-OH. The Lys(PEG-palmytoyl) group was incorporated into the targeted molecule via Fmoc-Lys(Dde)-OH addition into the peptide followed by deprotection with hydrazine. Fmoc-NH-PEG8-COOH was added via solid-phase peptide synthesis and deprotected followed by addition of palmitic acid, C15-COOH. The couplings were mediated by DIC/HOBt with a 3-fold excess of reagents. The resin bound peptides were treated with the TFA/TIS/H_2_O 95/2.5/2.5 (*v*/*v*/*v*) mixture to remove the acid sensitive protecting groups and were subsequently cyclized with BOP/DIPEA in DMF. The crude cyclic peptide was cleaved from the resin with HF/anisole 10/1 (*v*/*v*) and purified by semi-preparative HPLC (Kinetex 5 mM, 100 A pore size, Biphenyl/TMS, 10 × 250 mm column). Mobile phases were as follows A: 0.05% aq. formic acid, B: 0.05% formic acid in MeCN; 1–10 min/1–95% B gradient; retention time 5.94 min. The purity of the targeted molecule ASK2131 used for the subsequent in vitro and in vivo assays was 98.5%. HRMS (ESI) calculated for C_76_H_131_N_13_O_22_S: 1609.9252 (M+), found: 1609.9213. The molecular weight was 1611.

### 4.3. In Vitro Functional Assay Measuring OXTR

Functional activity of ASK2131 at the OXT receptor was assessed using a Ca^2+^ flux fluorometric imaging plate reader assay. In a typical experimental protocol, stably transfected Chinese hamster ovary (CHO) hOTR NFAT bLac cells were plated at a density of 10,000 cells/well in a 384-well plate in growth medium (Dulbecco’s modified Eagle’s medium + Glutamax (Thermo Fisher Scientific, Waltham, MA, USA), 10% dialyzed fetal bovine serum). The next day calcium-sensitive, fluo-4 dye (Invitrogen, Carlsbad, CA, USA) and probenecid (Invitrogen; 2.5 mM) was dissolved in 20 mL warm dye loading buffer (Hanks’ balanced salt solution with calcium and magnesium, 20 mM Hepes, pH 7.4) and added to each well. Plates were incubated for 60–120 min in a 37 °C cell incubator to allow cells to uptake the dye. The control peptide (OXT) and ASK2131 were diluted in Hanks’ balanced salt solution with calcium and magnesium, 20 mM HEPES, and 0.1% bovine serum albumin. After dye-loading, the cell plates were placed on the fluorometric imaging plate reader. A total of 15 mL of test compound was added to the 30 mL of dye on the cells, and intracellular Ca^2+^ was monitored for 90 s. Intracellular Ca^2+^ concentrations were calculated as the maximum increase over basal levels. Responses were normalized to control responses. Data were analyzed by curve fitting.

### 4.4. In Vitro Functional Assay Measuring V1aR

Evaluation of the agonist activity of OXT and ASK2131 at the human V1a receptor was conducted using stably transfected CHO cells by measuring the test article effect on cytosolic Ca^2+^ ion mobilization via a fluorimetric detection method. The cells were suspended in DMEM buffer (Invitrogen, Carlsbad, CA, USA) complemented with 0.1% FCS, then distributed in microplates at a density of 4.5 × 104 cells/well. The fluorescent probe (Fluo4 Direct, Invitrogen, Carlsbad, CA, USA) mixed with probenicid in HBSS buffer (Invitrogen, Carlsbad, CA, USA) complemented with 20 mM HEPES (Invitrogen; Carlsbad, CA, USA) (pH 7.4) was then added into each well and equilibrated with the cells for 60 min at 37 °C then 15 min at 22 °C. The assay plates were positioned in a microplate reader (CellLux, PerkinElmer, Waltham, MA, USA) followed by the addition of the test compound ASK2131, reference agonist or HBSS buffer (basal control), and the measurements of changes in fluorescence intensity which varies proportionally to the free cytosolic Ca^2+^ ion concentration. For stimulated control measurements, AVP at 1 μM was added in separate assay wells. The results were expressed as a percent of the control response to 1 μM AVP. The standard reference agonist was AVP, which was tested in each experiment at several concentrations to generate a concentration-response curve from which its EC_50_ value was calculated.

### 4.5. In Vitro Functional Assay Measuring V2R

Evaluation of the agonist activity of ASK2131 at the human V2 receptor expressed in stably transfected CHO cells was conducted by measuring its effects on cAMP production using the HTRF detection method. The cells were suspended in HBSS buffer (Invitrogen, Carlsbad, CA) complemented with 20 mM HEPES (pH 7.4), 0.01% BSA and 500 μM IBMX, then distributed in microplates at a density of 3.104 cells/well and incubated for 30 min at room temperature in the absence (control) or presence of the test compound or the reference agonist.

For stimulated control measurement, separate assay wells contained 1 nM AVP. Following incubation, the cells were lysed, and the fluorescence acceptor (D2-labeled cAMP) and fluorescence donor (anti-cAMP antibody labeled with europium cryptate) were added. After 60 min at room temperature, the fluorescence transfer was measured at λ_ex_ = 337 nm and λ_em_ = 620 and 665 nm using a microplate reader (Envision, Perkin Elmer, Waltham, MA, USA). The cAMP concentration was determined by dividing the signal measured at 665 nm by that measured at 620 nm (ratio).

The results were expressed as a percent of the control response to 1 nM AVP. The standard reference agonist was AVP, which was tested in each experiment at several concentrations to generate a concentration-response curve from which its EC_50_ value was calculated.

### 4.6. Preparation and Administration of Drug

In our hands, best and most consistent weight reducing effects were achieved using ASK2131 formulations with Hydroxypropyl-β-cyclodextrin (HP-CD) as an excipient. Specifically, ASK2131 stock solution was prepared using 10:1 (wt:wt) HP-CD (1541 g/mol) excipient to ASK2131 (1611 g/mol) in sterile ultra-pure H_2_O and gently agitated at 4 °C for >24 h. This excipient has been tested by our group previously to ensure that it does not elicit anorectic effects independently [16]. The stock was aliquoted and stored at −20 °C. Lyophilized oxytocin (Syntocinon) was purchased from Selleck Chemical (cat. # P1029; Houston, TX, USA) and dissolved in 1 mL if sterile ultra-pure H_2_O. The stock was aliquoted and stored at −20 °C. Stock solutions for both drugs were diluted to working solution in 0.9% normal saline solution and mixed gently prior to use. Normal saline solution was used as the vehicle control in all studies.

Vehicle, OXT, and ASK2131 were injected once daily at the start of the dark cycle using subcutaneous injection with a 1 cc 29G insulin syringe. The concentration of both OXT and ASK2131 solutions were adjusted so that dosing volume remained at 1 mL/kg.

### 4.7. Pharmacokinetics Study

Male lean Sprague Dawley rats (~300 g; *n* = 5 per group) were administered 300 nmol/kg dose of either ASK2131 or OXT via sc injection to assess absorption, peak concentration, and clearance. Serial blood samples (200 µL per time point) were obtained via tail nick using K_3_EDTA microvettes (mfg. # 20.1288.100; Sarstedt, Inc., Nümbrecht, Germany) prior to and 1, 2, 4, 6, 12, and 24-h post-drug administration. Whole blood samples were immediately centrifuged at 4 °C; separated plasma (~100 uL) was pipetted into micro centrifuge tubes, flash frozen on dry ice, and stored at −80 °C. Quantitative analysis plasma samples were performed by Alturas Analytics (Moscow, ID, USA) via peptide extraction from acidified plasma with acetonitrile followed by LC-MS/MS analysis.

### 4.8. Long-Term Drug Intervention, Blood Parameters, and Gene Expression

Male Sprague Dawley rats were fed 60% HF diet (formula: D12492; Research Diets, Inc; New Brunswick, NJ, USA; 5.21 kcal/g) for 16 weeks prior to the start of the study. Rats were then singly housed in BioDAQ cages (Research Diets, Inc.; New Brunswick, NJ, USA) and allowed to acclimate to their new environment for 10 days. Baseline measures of body weight, food intake and water intake were taken for one week to balance the groups and create feeding pairs. Each group (*n* = 6 animals/group) averaged 714.47 ± 81.60 g at baseline (Table 3).

Effects of ASK2131 were compared to a vehicle-treated group, OXT-treated group, as well as a pair-fed control group. Three days of baseline measures were taken before subcutaneous treatment administration began on day 4 of the study. Assessments were made to body weight, food intake and water intake. ASK2131 was administered at 50 nmol/kg throughout the entire study period. OXT doses were titrated during the first 6 days of the study. OXT-treated animals were started at 50 nmol/kg dose for two days, followed by 150 nmol/kg, then 300 nmol/kg, until ending at 600 nmol/kg for the remainder of the study.

Cardiac puncture was used to collect blood for analysis 2 h post final injection. Commercially available enzyme-linked immunosorbent assays (ELISA) were used for quantitative assessment of leptin (cat. # EZRL-83K; MilliporeSigma, Burlington, MA, USA), insulin (cat. # EZRMI-13K; MilliporeSigma, Burlington, MA, USA), and adiponectin (cat. # EZRAPD-62K MilliporeSigma, Burlington, MA, USA). Serum samples were diluted 1:500 for the adiponectin ELISA using the sample diluent provided with the kit. All ELISAs were completed following the manufacturers recommendations. Levels of glucose, cholesterol (total, HDL, and calculated LDL), triglycerides, ALT, and AST were determined in the serum on a Modular P chemistry analyzer (Roche Diagnostics, Risch-Rotkreuz, Switzerland) by the University of Washington Nutrition and Obesity Research Center’s Analytic Core (Seattle, WA, USA).

IWAT and IBAT were collected from the DIO rats treated with vehicle, oxytocin, ASK2131, and the pair-fed to the ASK2131. IWAT and IBAT were homogenized using a TissueLyser LT (Qiagen, Hilden, Germany). RNA was extracted from the tissue lysates using the RNeasy Lipid Mini Kit (mfg. # 74804; Qiagen, Hilden, Germany) and reverse transcribed using the high-capacity cDNA kit (mfg. # 4368814; Applied Biosystems, Waltham, MA, USA). qRT-PCR was completed on the mRNA samples using the Applied Biosystems StepOnePlus System (Applied Biosystems, Waltham, MA, USA). All samples were normalized to the cycle threshold value of 18S mRNA. The following TaqMan^®^ probes (Applied Biosystems, Waltham, MA, USA) were used in this study: 18S (cat. # Hs99999901_s1), UCP1 (cat. # Rn00562126_m1), and Cox8b (cat. # Rn00562884_m1). Analysis was completed using the delta CT method.

A secondary analysis was performed comparing the first 22 days of ASK1476 and OXT treatment from our reported experiment [16] with ASK2131 and OXT in the current 22-day experiment. Analyses were limited to body weight change and differences in cumulative food intake.

### 4.9. Dose Escalation Study

Male DIO Sprague Dawley rats (*n* = 4) were fed 60% HF diet (formula: D12492; Research Diets, Inc., New Brunswick, NJ, USA; 5.21 kcal/g) for 16 weeks prior to the start of the study. Rats were changed to being singly housed and allowed to acclimate for at least one week prior to the start of this study. Animals averaged 844.15 ± 99.95 g at baseline. Baseline measures of body weight, food intake, and water intake were taken for 6 days before drug administration began. Doses for the dose escalation study included 1, 2, 5, 10, 20, 50, and 100 nmol/kg. Body weight, food and water intake were assessed just prior to the start of the dark cycle.

### 4.10. Statistical Analyses

Statistical analyses were performed using GraphPad Prism 9 Software (La Jolla, CA, USA. Two-way ANOVAs were used for comparison of repeated measures with Šídák’s multiple comparison test for post hoc analyses. For cross-sectional comparisons of normally distributed measures taken at only one time point, an ordinary one-way ANOVA was used when group variances are assumed equal, and Welch’s ANOVA was used for unequal variances. Post hoc comparisons between groups were made using Šídák’s multiple comparison test. Skewed data were analyzed using a nonparametric test, i.e., Kruskal–Wallis test assuming equal variances or Brown-Forsythe ANOVA assuming unequal variances; an F-test was used to compare variances between groups and post hoc comparisons were made using Dunn’s or Dunnett’s T3 multiple comparison tests as appropriate. In all instances, a *p* < 0.05 was considered significant. All values are represented as mean (SEM).

## Figures and Tables

**Figure 1 ijms-23-11249-f001:**
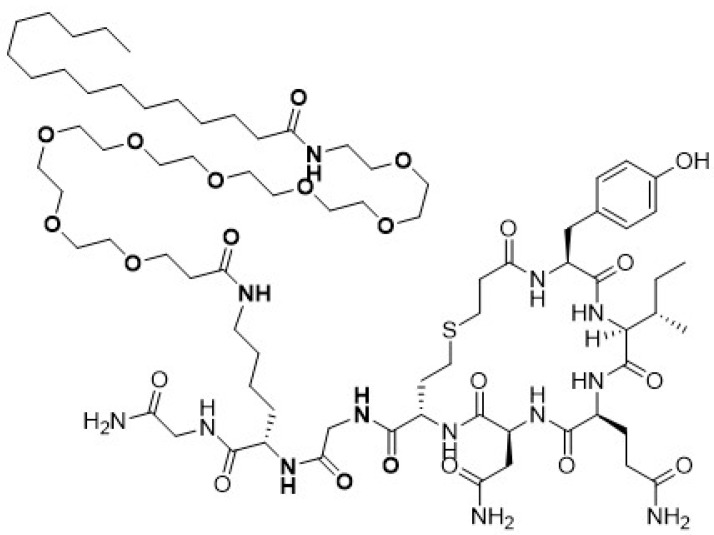
Structure of novel long-acting oxytocin analogue ASK2131. The native oxytocin peptide was modified with a substitution of the Leu8 to a Lys appended with a polyethylene glycol space and a palmitoyl group and with a substitution of Gly for the Pro7 to enhance selectivity for the OXT receptor to create the prior peptide ASK1476. The design of ASK1476 had been further enhanced by replacing its disulfide bridge with a thioether isostere and removing the Cys amino group, to create ASK2131, a novel OXT receptor analog.

**Figure 2 ijms-23-11249-f002:**
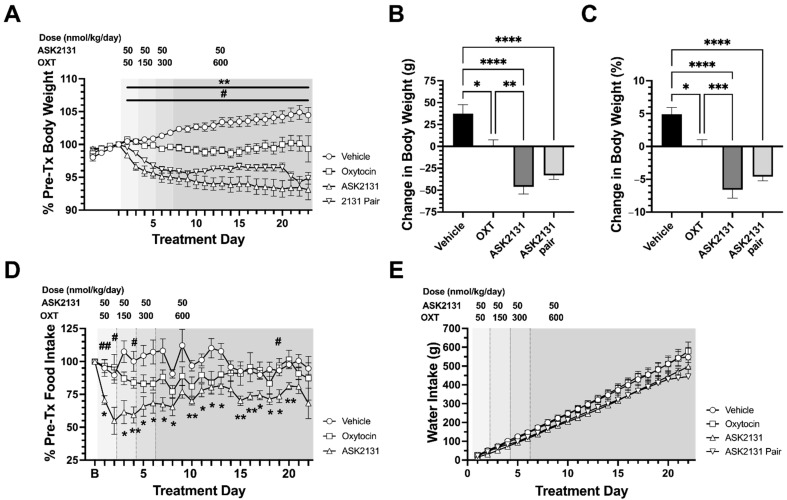
Changes in body weight, food intake, and water intake were tested during 22-day treatment with ASK2131 vs. pair-feeding to ASK2131 vs. oxytocin (OXT) vs. vehicle in diet-induced obese (DIO) male Sprague Dawley rats which were matched based on baseline food intake and body weight gain trajectory. Body weight was reduced in all treatment groups with strongest reductions in percent reductions of body weight (**A**,**B**), and total body weight (**C**) in rats treated with ASK2131, followed by pair-fed to ASK2131. Daily food intake was reduced during the whole duration of 22-day treatment with ASK2131 in comparison to vehicle (**D**). Water intake was comparable between all treatment groups (**E**). Figure 2A,D: Data were analyzed by repeated measure two-way ANOVA and Šídák’s correction for multiple comparisons for comparing treatment groups vs. vehicle. ^#^/^##^
*p* < 0.05/0.01 ASK2131 vs. OXT. */** *p* < 0.05/0.01 ASK2131 vs. vehicle. Figure 2B,C: Analysis by one way ANOVA followed by Šídák’s multiple comparisons test. * *p* < 0.05, ** *p* < 0.01, *** *p* < 0.001, **** *p* < 0.0001.

**Figure 3 ijms-23-11249-f003:**
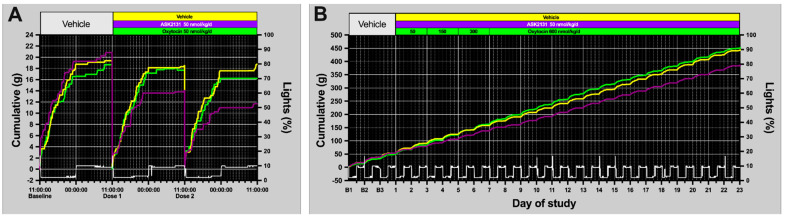
Analysis of food intake pattern of continuously automatically recorded food intake of singly housed rats in BioDAQ cages. Strong reductions of food intake were noted already at days 1 and 2 during treatment with ASK2131, but not to OXT compared to baseline, both drugs at 50 nmol/kg (**A**). Cumulative food intake throughout the whole experiment was more reduced in response to ASK2131 than to oxytocin, even when doses were increased for oxytocin every 2 days up to 600 nmol/kg while the dose for ASK2131 was kept at 50 nmol/kg/d (**B**). Green—Vehicle; yellow—Oxytocin; purple—ASK2131; White—ambient light data recorded by BioDAQ system.

**Figure 4 ijms-23-11249-f004:**
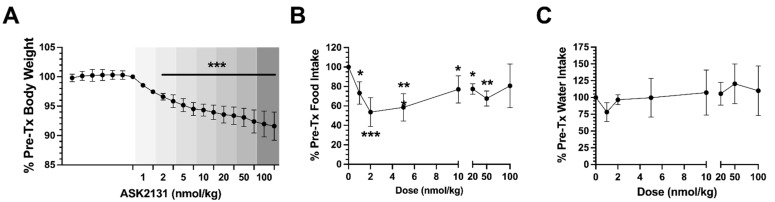
Changes of weight (**A**), food intake (**B**), and water intake (**C**) in DIO rats treated with ASK2131 after 6 days of baseline measures, starting dose 1 nmol/kg day, then increase to 2, 5, 10, 20, 50, and finally 100 nmol/kg every 2 days. The low dose of ASK2131 was effective in significantly reducing body weight and food intake, higher doses did not result in greater reductions of food intake and body weight. Analysis by Two-Way ANOVA test with Šídák’s multiple comparisons test. * *p* < 0.05 vs. baseline, ** *p* < 0.01 vs. baseline, *** *p* < 0.0001 vs. baseline.

**Table 1 ijms-23-11249-t001:** Receptor pharmacology profile of both oxytocin (OXT) and ASK2131 against OXTR, human vasopressin 1a receptor (hV1aR) and human vasopressin 2 receptor (hV2R).

Compound	hOXTR, EC_50_ (nM)	hV_1a_R, EC_50_ (nM)	hV2R, EC_50_ (nM)	Ratio hOXTR/hV2R
OXT [1]	2.3	10	7.3	0.32
ASK2131	1.1	NA (>1000)	0.36	3.06

**Table 2 ijms-23-11249-t002:** Pharmacokinetic parameters for oxytocin (OXT) [2] and ASK2131 following subcutaneous (sc) administration to rats and mice.

Compound Route	CL_plasma_, mL/min/kg	V_d_, L/kg	C_max_/Free C_max_, nM	T_max_, Hour	T_1/2_, Hour
OXT, sc (rat) [18]	17.1	0.085			0.12
OXT, sc (mouse) [18]			216/191	0.25	0.5
ASK2131, sc (rat)	1.20	0.80	276/0.5	4.00	2.30

**Table 3 ijms-23-11249-t003:** Pre- and post-treatment body measures, and post treatment serum levels.

	Vehicle	Oxytocin	ASK2131	ASK2131 Pair
*n*=	6	6	6	6
Pretreatment body weight (g)	762 ± 64	719 ± 18	723 ± 38	724 ± 43
Posttreatment body weight (g)	799 ± 69	719 ± 20	676 ± 42	691 ± 41
% change in body weight	4.9 ± 1.1	0.1 ± 1 ^b^	−6.6 ± 1.3 ^c,d^	−4.6 ± 0.6
Glucose (mg/dL)	232 ± 8	203 ± 13	237 ± 12	256.± 13
Triglycerides (mg/dL)	190 ± 10	145 ± 7	182 ± 30	128 ± 7
Cholesterol (mg/dL)	102 ± 8	76 ± 3	67 ± 6 ^b^	73 ± 6 ^a^
HDL (mg/dL)	28 ± 3	25 ± 2	21 ± 2	23 ± 2
Calculated LDL (mg/dL) *	31 ± 5	22 ± 2	15 ± 5	24 ± 5
ALT (U/L)	44 ± 13	40 ± 5	26 ± 2	28 ± 3
AST (U/L)	117 ± 30	128 ± 22	73 ± 8	100 ± 8
Leptin (mg/dL)	55.5 ± 11.1	54.9 ± 7.9	39.0 ± 9.1	37.8 ± 8.0
Insulin (ng/mL)	5.6 ± 1.5	7.2 ± 2.1	6.7 ± 1.0	10.2 ± 2.6
Adiponectin (ng/mL)	76.2 ± 2.7	60.8 ± 3.9	62.4 ± 7.3	48.2 ± 6.4

^a^: *p* < 0.05 vs. vehicle ^b^: *p* < 0.01 vs. vehicle ^c^: *p* < 0.001 vs. vehicle ^d^: *p* < 0.001 vs. OXT. Data expressed as mean ± SEM. Primary Analysis for normal distribution: One-Way ANOVA test assuming equal variances or Welch’s ANOVA for unequal variances; Šídák’s multiple comparisons test was used for post hoc analyses in both instances. For non-normally distributed data: Kruskal–Wallis test with Dunn’s multiple comparison test assuming equal variances or Brown-Forsythe ANOVA test with Dunnett’s T3 multiple comparison test for unequal variances. Serum measures were only available for *n* = 4 of the Oxytocin treated animals. * Calculated LDL = Cholesterol − (HDL + Triglycerides/5).

## Data Availability

The raw data supporting the findings of this article will be made available by the corresponding author upon reasonable request.

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
