# Peer review of "Novel Long-Acting Oxytocin Analog with Increased Efficacy in Reducing Food Intake and Body Weight"

_ijms, 2022, doi:10.3390/ijms231911249_

Round 1

Reviewer 1 Report

The present manuscript examined the impact of an oxytocin analog on diet-induced obesity outcomes in a rat model. This work built on promising previous findings that used the natural form of oxytocin, which has some drawbacks, short half-life being a major one. This new approach offers a longer-lasting solution to activating oxytocin receptors. Overall, I found this study well designed and well conducted. The results suggest a promising advancement in pharmacological treatment of diet induced obesity. The manuscript does not shy away from its weaknesses, such as only examining males. However, I did not identify any critical weaknesses that would prevent this work from being published.

Discussion Line 223 – Was fluid imbalance examined the present work? My understanding was that it was only assessed in terms of fluid consumption. I think this is an area that deserved further investigation and assurance in the future, though that can wait for now. At present, the language of the manuscript should just be careful about sticking to exactly what was assessed.

This study could have been strengthened by the inclusion of more measures relevant to metabolism such as locomotor activity. For instance, what does the magnitude of decrease in food consumption as compared to the magnitude of weight loss tells us? Would we expect a 11.5% decrease in body weight after a 28% decrease in food intake -or does this imply compensatory changes in locomotor activity, metabolic rate, etc.?

The first mention of UCP1 and Cox8b mRNA comes in the Discussion. These measures and results should be introduced earlier.

The addition of the lipid palmitoyl group to enhance plasma protein binding of the peptide and thereby increase half-life, struck me as interesting. Would such binding to plasma proteins not preclude the peptide from binding to the relevant oxytocin receptors? This is more for my personal curiosity than the soundness of the manuscript. The question of plasma protein binding is relevant to ongoing debates about plasma oxytocin assay.

Reviewer 2 Report

The research article by Elfers et al reported the design, pharmacokinetics, receptor selectivity of a novel oxytocin agonist, ASK2131. Previously author reported ASK1476, as an oxytocin (OXT) agonist, with improved in vivo stability. Their previous article claims that daily one time subcutaneous administration of ASK1476 reduced food intake body weight comparably to human studies which currently require 4x daily intranasal administration of OXT. However, there were many adverse effects occurring at higher doses of drug administration that motivated author of the current paper to modify ASK1476. They call modified ASK1476 as ASK2131.  They introduced 2 key modification by first replacing disulfide bridge -S-S- of ASK1476 with the respective thioether isostere and the Cys amino group was eliminated and secondly, they replaced 7Pro with 7Gly to improve V1aR selectivity and 8Leu was changed to 8Lys followed by modification of terminal NH2  with polyethylene glycol spacer and a lipid palmitoyl group. This OXT analog is a full agonist at OXT receptor. In this paper they reported its efficacy in reducing caloric intake and body weight by testing it on diet- induced obese (DIO) Sprague-Dawley rats. On analysis author found that  ASK2131 is a longer-lasting OXT analog with improved pharmacokinetics compared to OXT (T1/2: 2.3 vs. 0.12h).It has improved potency leading to a profound reduction in body weight partly explained by reduced food intake. Overall, the finding is a considerable progression to previous paper and will be valuable to the readers of IJMS, however, there is a concern that need to be addressed for research article to be considered for publication.

Comment: Throughout the results section author have drawn comparisons between OXT administration and ASK2131 but no comparison is highlighted between ASK2131 and its precursor ASK1476. Although I understand that findings of ASK1476 would be reported in previous paper, drawing the comparison in the current paper as OXT VS ASK1476 VS ASK2131 will be  more  informative and easy for the reader to gauge the advantages of such modification.
